# Automatic Detection and Characterization of Coronary Artery Plaque and Stenosis using a Recurrent Convolutional Neural Network in Coronary CT Angiography

Majd Zreik[1,*], Robbert W. van Hamersvelt[2], Jelmer M. Wolterink[1], Tim Leiner[2], Max A. Viergever[1], and Ivana Išgum[1]

[1]Image Sciences Institute, University Medical Center Utrecht, Utrecht, The Netherlands
[2]Department of Radiology, University Medical Center Utrecht and Utrecht University, Utrecht, The Netherlands
* Corresponding author, `m.zreik@umcutrecht.nl`

## Abstract

Different types of atherosclerotic plaque and varying grades of stenosis lead to different management of patients with obstructive coronary artery disease. Therefore, it is crucial to determine the presence and classify the type of coronary artery plaque, as well as to determine the presence and the degree of a stenosis. The study includes consecutively acquired coronary CT angiography (CCTA) scans of 131 patients. In these, presence and plaque type in the coronary arteries (no plaque, non-calcified, mixed, calcified) as well as presence and anatomical significance of coronary stenosis (no stenosis, non-significant, significant) were manually annotated by identifying the start and end points of the fragment of the artery affected by the plaque. To perform automatic analysis, a multi-task recurrent convolutional neural network is utilized. The network uses CCTA and coronary artery centerline as its inputs, and extracts features from the region defined along the coronary artery centerline using a 3D convolutional neural network. Subsequently, the extracted features are used by a recurrent neural network that performs two simultaneous multi-label classification tasks. In the first task, the network detects and characterizes the type of the coronary artery plaque. In the second task, the network detects and determines the anatomical significance of the coronary artery stenosis. The results demonstrate that automatic characterization of coronary artery plaque and stenosis with high accuracy and reliability is feasible. This may enable automated triage of patients to those without coronary plaque, and those with coronary plaque and stenosis in need for further cardiovascular workup.

## 1 Introduction

Coronary artery disease (CAD) is the most common type of heart disease (1). Obstructive CAD occurs when atherosclerotic plaque builds up in the wall of the coronary arteries. This may lead to stenosis, i.e. narrowing or occluding of the coronary artery lumen, limiting blood supply to the myocardium, and potentially leading to myocardial ischemia. Atherosclerotic plaque can be classified according to its composition into calcified plaque, non-calcified plaque, and mixed plaque, i.e plaque containing calcified and non-calcified components. As different types of plaque and varying grades of luminal narrowing, i.e. arterial stenosis, lead to different patients management strategies, it is crucial to detect and characterize coronary artery plaque and stenosis (2).

1st Conference on Medical Imaging with Deep Learning (MIDL 2018), Amsterdam, The Netherlands.

Coronary CT angiography (CCTA) is a well-established modality for identification of patients with suspected CAD. It allows noninvasive detection and characterization of coronary artery plaque and grading of coronary artery stenosis. To day, these tasks are typically performed by visual assessment (2), or semi-automatically by first utilizing lumen segmentation and thereafter defining the presence of a plaque or a stenosis (3). However, the former suffers from substantial interobserver variability, even when performed by experienced experts, while the latter is dependent on the coronary artery lumen segmentation that is typically time-consuming and cumbersome. Most of the commercially available software packages for coronary artery lumen segmentation require substantial manual interaction, especially in images with excessive atherosclerotic plaque or imaging artefacts (3).

The amount of calcium in coronary arteries is a strong and independent predictor of cardiovascular events. In recent years, several methods have been developed to automatically segment and quantify calcifications in the coronary arteries in cardiac CT and CCTA scans (e.g. (4; 5)). Most often, these methods employ machine learning and analyze CT scans reconstructed to axial slices. Typically good performance approaching the level of an expert is achieved (4). On the contrary, detection and quantification of non-calcified coronary plaque using CCTA has not been extensively investigated. Unlike methods detecting calcifications, non-calcified plaque is detected in straightened multi-planar reformatted (MPR) images. MPR allows better visualization of the arterial lumen and allows identification of difficult to delineate non-calcified plaque in the arterial wall. Previously proposed methods perform manual or semi-automatic thresholding on CT values (Hounsfield units) to detect and quantify non-calcified plaque (6). Typically, these methods require substantial manual interactions by experts.

Given the importance of stenosis detection and grading, a number of methods have been developed to (semi-)automatically detect and grade coronary artery stenosis in CCTA (3). These methods either segment arterial lumen (7; 8), or utilize machine learning approach to analyze the vicinity of the coronary artery centerline (9; 10; 11) to detect and quantify stenosis. Machine learning algorithms design a number of features computed along a region of interest around the centerline of an artery to describe local image intensity and arterial geometry. Subsequently, they use a supervised classifier to detect and quantify stenosis. Segmentation algorithms delineate coronary artery lumen to subsequently detect and quantify stenosis by analyzing local changes and anomalies in the diameter of the delineated artery. A number of methods have been evaluated in the Rotterdam Coronary Artery Stenoses Detection and Quantification Challenge[1] (3) to (semi-)automatically detect and grade coronary artery stenosis in CCTA. Best performance was achieved by the following three methods. Shahzad et al. (12) first extracted the centerline of the artery and subsequently employed a graph cut approach and robust kernel regression to segment the arterial lumen. Thereafter, to detect and grade coronary stenosis, the diameter of the segmented lumen was compared with the expected diameter of a healthy lumen. The expected diameter of the healthy lumen was estimated by regression of the diameters in the adjacent coronary artery lumen. Wang et al. (13) employed a level-set model to separately segment the inner and the outer arterial walls. Then, a comparison between these arterial walls profiles enabled the detection and grading of a stenosis. Similarly, Broersen et al. (14) first segmented the arterial lumen and found vessel wall contours and thereafter, detected regions potentially containing stenosis based on deviations from the normal lumen using a regression model.

In this work, we present a method to automatically detect and characterize the type of the coronary artery plaque, as well as to detect and determine the anatomical significance of the coronary artery stenosis in CCTA scans. Our method does not require segmentation of the coronary artery lumen but instead, it relies on the coronary artery centerlines. First, the centerlines were extracted using our previously developed method (15). Together with CCTA image, the centerlines were used as input to the presented algorithm. Thereafter, a multi-task recurrent convolutional neural network (RCNN) is employed to analyze the vicinity of a number of points along the extracted centerline. The RCNN consists of a 3D convolutional neural network (CNN) that extracts image features from a volume centered around each centerline point, and a recurrent neural network (RNN) that analyzes the features extracted from the sequence of analyzed volumes. Finally, two classification tasks are simultaneously performed. First, the network detects and characterizes the type of the coronary artery plaque. Second, the network detects and determines the anatomical significance of the coronary artery stenosis. To the best of our knowledge, we are the first to propose a method for jointly performing both plaque and stenosis characterizations. Moreover, unlike other automatic methods, we omit an intermediate lumen segmentation task, thereby preventing potential propagation of errors.

---

[1]http://coronary.bigr.nl/ stenoses/

## 2   Data

### 2.1   Patient and image data

This study includes retrospectively collected CCTA scans of 131 patients (age: $58.8 \pm 9.4$ years, 105 males) acquired in our hospital between 2012 and 2016. The Institutional Ethical Review Board waived the need for informed consent. All CCTA scans were acquired using an ECG-triggered step-and-shoot protocol on a 256-detector row scanner (Philips Brilliance iCT, Philips Medical, Best, The Netherlands). A tube voltage of 120 kVp and tube current between 210 and 300 mAs were used. For patients $\leq 80$ kg contrast medium was injected using a flow rate of 6 mL/s for a total of 70 mL iopromide (Ultravist 300 mg I/mL, Bayer Healthcare, Berlin, Germany), followed by a 50 mL mixed contrast medium and saline (50:50) flush, and a 30 mL saline flush. For patients $> 80$ kg the flow rate was 6.7 mL/s and the volumes of the boluses were 80, 67 and 40 mL, respectively. Images were reconstructed to an in-plane resolution ranging from 0.38 to 0.56 mm, with 0.9 mm slice thickness and 0.45 mm slice increment.

In each CCTA image, luminal centerlines of coronary arteries were extracted using the method previously developed by Wolterink (15). The method requires manual placement of a single seed point in the artery of interest, after which the arterial centerline is extracted between the ostium and the most distal point as visualized in the CCTA image. In total, centerlines for 440 arteries were extracted. Using the extracted centerlines, a 0.3 mm isotropic straightened MPR image was reconstructed for each artery and used for further analysis.

### 2.2   Reference standard

To define a reference standard for atherosclerotic plaque and coronary artery stenosis, straightened MPR images showing coronary artery were used. Only arteries with a diameter greater than 1.5 mm were annotated. Plaque type and anatomical significance of the stenosis were manually annotated by an expert using custom-built software created with the MeVisLab[2] platform and following the society of cardiovascular computed tomography (SCCT) guidelines for coronary artery disease reporting system (Fig. 1) (2). For each plaque, the expert marked its start and end points, its type (non-calcified, mixed (i.e. with $< 50\%$ calcified volume), calcified (i.e. with $\geq 50\%$ calcified volume)), and the anatomical significance of the stenosis caused by the plaque (no stenosis, non-significant with narrowing $< 50\%$ in luminal diameter, significant with narrowing $\geq 50\%$ in luminal diameter). The significance of a stenosis was determined by visual estimation of the maximal grade of luminal narrowing cause by a plaque. Note that the plaque was not segmented, but the part of the artery containing the plaque was identified. The expert also annotated a number of plaque-free and stenosis-free parts of the arteries in different patients. In further text we refer to the annotated and automatically analyzed parts of the arteries as fragments. Note that these are not anatomically defined coronary artery segments.

The data set contained 1073 manually labeled arterial fragments. These included manually annotated fragments with 21 non-calcified, 150 mixed and 304 calcified plaques with non-significant stenosis. Furthermore, there were fragments with 27 non-calcified, 91 mixed and 40 calcified plaques that caused significant stenosis. 440 plaque-free and stenosis-free fragments of the arteries were defined.

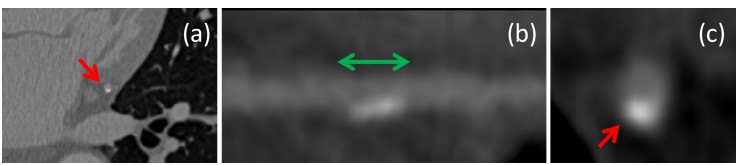

*Figure 1: Axial (left) and straightened MPR image with longitudinal view (center), and cross-sectional view (right) showing coronary artery in CCTA. This artery contains a lesion (spanning the green arrow) labeled as containing calcified plaque with non-significant stenosis. Red arrows indicate the location of this plaque in other views.*

---

[2]http://www.mevislab.de

# 3 Method

To detect and characterize the type of the coronary artery plaque, as well as to detect and determine the anatomical significance of the coronary artery stenosis, an RCNN is designed. Recently, RCNNs have been successfully used for video recognition and description (e.g. (16)) object recognition (e.g. (17)), speech modeling (18), and recently also in medical image analysis (e.g. (19)). RCNNs typically connect a CNN with an RNN in series to analyze a sequential input. The CNN extracts image features for each input of the sequence independently (e.g. frame in a video clip, word in a sentence, cardiac phase in cardiac cycle), and these extracted features are then fed to the RNN that analyzes the relevant sequential dependencies in the whole sequence.

In this work, reference annotation was performed for fragments of the coronary arteries and not for each voxel in the arterial wall or each cross-section of the arterial lumen. Given that the appearance of plaque along the whole fragment is important for characterization of its type and for determination of stenosis significance, we propose to analyze fragments of the artery as sequences of volumes along the coronary artery centerline rather than to analyze these volumes independently. For each analyzed sequence, the network detects coronary artery plaque and characterizes the type of the detected plaque (no plaque, non-calcified, mixed, calcified), and detects coronary artery stenosis and determines the anatomical significance of the detected stenosis (no stenosis, non-significant, significant). The input of the network consists of a sequence, spanning the analyzed fragment. The sequence consists of a number of 3D cubes extracted along the coronary artery centerline in the MPR spanning one fragment. Each of these cubes are separately analyzed by a 3D CNN that extract features. Thereafter, an RNN is deployed to analyze and accumulate the features extracted by the CNN from all cubes into a single vector of features representing the entire sequence. Finally, the feature vector is fed into two softmax classifiers to perform the two tasks of plaque and stenosis characterization.

## 3.1 Network architecture

An illustration of the proposed network is shown in Fig. 2. The input of the network is a sequence of a maximal length of 25 cubes of $25 \times 25 \times 25$ voxels with stride of 5 voxels along the coronary artery centerline, extracted from the MPR. Each cube is analyzed by a 3D CNN. The CNN consists of three convolutional layers with kernels of $3 \times 3 \times 3$, with 32, 64, 128 filters, respectively. Each convolutional layer is followed by $2 \times 2 \times 2$ max-pooling layer and batch normalization (20). The features extracted by the CNN are fed to the RNN, which models the sequential information. The RNN consists of 2 layers of 64 Gated Recurrent Units (GRUs) (21) each. Rectified linear units (ReLU) (22) are used in both CNN and RNN layers as activation functions. The output of the last layer of the RNN is fed into two separated multi-label softmax classifiers. The first classifier has four output units for detection of plaque and characterization of its types (no-plaque, non-calcified, mixed, calcified). The second classifier has three output units for detection of stenosis and determination of its anatomical significance (no-stenosis, non-significant, significant). The network has a total number of 340,295 parameters.

## 3.2 Training strategy

During training and validation, manually annotated reference fragments were used. The network was trained using mini-batches containing labeled fragments. In the dataset, the distribution of plaque types and stenosis grades is very unbalanced. To overcome this, a stratified random data sampling was performed for the training. Each training iteration included two distinct mini-batches. One mini-batch contained fragments balanced with respect to their plaque classes regardless of the stenosis significance. A second mini-batch contained fragments balanced with respect to the stenosis classes of the plaque regardless of the plaque type.

Additionally, several data augmentation techniques were utilized to increase the training set size. First, to make the network invariant to rotations around the artery centerline, random rotations between 0 and 360 degrees around the coronary artery centerline were applied to all cubes of a sequence. Second, to make the network invariant to slight inaccuracies in manual annotations of the points defining the fragment, a sequence of a fragment is varied by randomly choosing centers of cubes with a stride of 5 voxels with a uniform random shift between $\pm 3$ voxels along the MPR centerline. Third, to make the network robust to possible inaccuracies in the extraction of the coronary artery centerline, the center of each cube was randomly shifted around its origin by up to $\pm 2$ voxels, in any direction.

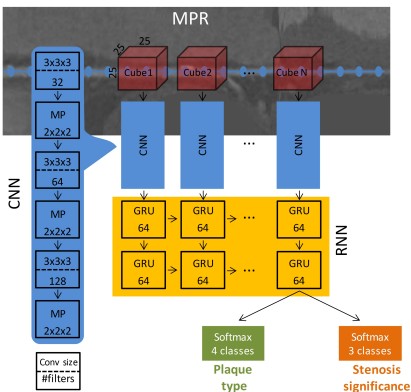

*Figure 2: Overview of the proposed network. An MPR is obtained using the artery centerline points (blue dotted line). The input of the network is a sequence of cubes extracted from the MPR, along the artery centerline. A CNN extracts features out of each $25 \times 25 \times 25$ voxels cube, then an RNN processes the entire sequence using GRUs. The output of the RNN is fed into two softmax classifiers to simultaneously characterize plaque and stenosis.*

## 3.3 Testing

Unlike in the training phase, during testing, the start and end points of a fragment are not available. Therefore, all points along the coronary centerline were classified and labeled by the network. This was done by feeding the network a fixed-length sequence centered around each centerline point. The fixed-length sequence consisted of 5 cubes with a stride of 5 voxels extracted along the coronary artery centerline. These parameters were optimized in preliminary experiments on the training and validation sets. The output probabilities are then assigned to the center of the evaluated sequence. Then, labels for each centerline point were determined by the label with the highest probability in each task separately.

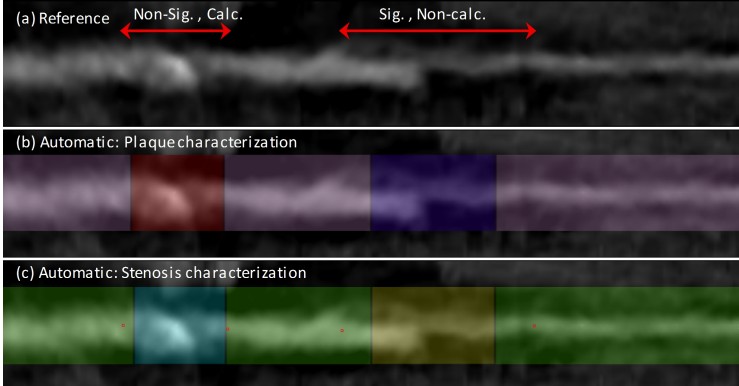

*Figure 3: An example of a result for an entire artery. (a) MPR view of an artery with reference standard. Each arrow represents a manually annotated single fragment, its boundaries and the reference labels. Sig. = Significant stenosis, Non-Sig. = Non-significant stenosis. Calc. = Calcified plaque. Non-calc. = Non-calcified plaque. (b) Automatically predicted plaque labels (no-plaque, non-calcified and calcified: pink, blue and red overlays, respectively). (c) Automatically predicted stenosis labels (no-stenosis, significant and non-significant: green, yellow and light blue, respectively). For illustration purposes only, all overlays were extended to a $25 \times 25$ voxels around the artery centerline.*

## 3.4 Evaluation

Performance of the network was evaluated on fragment-, artery- and patient-basis. For fragment-based evaluation, only the predicted labels along the centerlines that fall within the manually annotated fragment boundaries were considered. For artery-based evaluation, all predicted labels along the

complete artery centerline were taken into account. For patient-based evaluation, all predicted labels along the extracted centerlines of all arteries of a patient were taken into account.

Automatically determined fragment is considered to be a true positive in detection of plaque type or stenosis significance when it overlaps with the manually annotated fragment of the same label. Automatically detected stenosis significance for an artery or patient is considered true positive if the automatically detected fragments of the artery or patient overlap with the fragments of the same label defined by the reference annotations. On contrary, a fragment is considered to be true positive in the detection of plaque absence (no plaque label) only when no point along the fragment has any plaque. Similarly, the fragment, artery or patient is considered to be true positive in the detection of stenosis absence (no stenosis label) only when no stenosis was detected at any point along the centerline of the fragment, artery or patient's arteries, respectively.

The evaluation of plaque detection and characterization was performed on a fragment-level only. The average accuracy of the prediction, i.e. the percentage of correctly labeled fragments, was computed. To account for both false negatives and false positives and to assess the overall performance, the unweighted average of F1 score was computed. This was done for each label separately, and computing the unweighted mean across all labels. Given the multiple categories of the plaque labels, Cohen's $\kappa$ metric was used to measure the reliability between the predicted plaque labels and the reference standard. Note that, for plaque detection and characterization, the unweighted Cohen's $\kappa$ was used.

The evaluation of stenosis detection and characterization was performed on fragment-, artery- and patient-levels. The average accuracy and the unweighted average F1 score were computed to assess the overall agreement for predicting the stenosis labels. Given the grading of the stenosis, Cohen's linearly weighted $\kappa$ metric was used to measure the reliability between the predicted stenosis labels and the reference standard.

# 4 Experiments and results

From the available data set consisting of 131 studies, CCTA scans of 79 (60%) and 19 (15%) patients were randomly chosen for training and validation, respectively. The scans of the remaining 33 (25%) patients were used for testing the method. All CNN and RNN hyperparameters were determined in preliminary experiments using the training and validations scans only.

During training and validation, sequences of cubes were extracted from fragments containing manual annotations only. For each fragment, a sequence of cubes, spanning the entire length of the fragment, was extracted from the MPR volume along the artery centerline. The maximal length of a sequence was set to 25 cubes, which corresponded to the maximal length of a fragment in the training data. Shorter sequences, spanning shorter fragments, were zero-padded to the maximal length. The categorical cross-entropy was used as loss function of each softmax classifier. The loss of the network was defined as the average of the two individual losses. Mini-batches of 30 sequences were used to minimize the loss function with Adam optimizer (23) with learning rate 0.0001. L2 regularization was used with $\gamma = 0.001$ for all layers. A random dropout (24) of 50% was applied in each recurrent layer to prevent overfitting. The training process was performed until the network converged. To evaluate the effect of data augmentation during training, an identical network was trained without any augmentation. validation accuracy during training process and the affect of data augmentation are both illustrated Fig. 1 in the Appendix.

During testing, prediction was performed for each point along the length of the coronary artery centerline (Section 3.3). An example of an artery with predicted labels for both classification tasks is shown in Fig. 3. An example of automatically predicted probabilities for both softmax classifiers providing more insight into the classification output is shown in the Appendix in Fig. 3. Table 1 lists the confusion matrix obtained for plaque detection and characterization. For plaque detection (plaque vs. no plaque) the method achieved an accuracy of 0.86 and an F1 score of 0.85. For plaque detection and characterization (no plaque, non-calcified, mixed, calcified) the method achieved an accuracy of 0.72, an average F1 score of 0.61 and a $\kappa$ of 0.60. For plaque characterization (non-calcified, mixed, calcified) the method achieved an accuracy of 0.76, an average F1 score of 0.66 and a $\kappa$ of 0.54.

Table 2 lists the confusion matrices obtained for detection and determination of stenosis anatomical significance. At a fragment-level, for detection of the significant stenosis (significant stenosis vs. no

*Table 1: Confusion matrix showing fragment-based results of detection and characterization of plaque by classification into no-plaque, non-calcified plaque, mixed plaque, and calcified plaque.*

| Fragment-based | Automatic | | | |
|---|---|---|---|---|
| | No plaque | Non-calcified | Mixed | Calcified |
| No-plaque | **77** | 8 | 1 | 9 |
| Non-calcified | 5 | **4** | 1 | 0 |
| Mixed | 5 | 4 | **22** | 22 |
| Calcified | 7 | 2 | 3 | **73** |

(Reference)

*Table 2: Confusion matrices showing fragment-, artery-and patient-based results for detection and characterization of the stenosis by classification into no-stenosis, non-significant and significant stenosis.*

| Fragment-based | Automatic | | |
|---|---|---|---|
| | No stenosis | Non-significant | Significant |
| No-stenosis | **78** | 15 | 2 |
| Non-significant | 14 | **99** | 2 |
| Significant | 1 | 12 | **20** |

(Reference)

| Artery-based | Automatic | | |
|---|---|---|---|
| | No stenosis | Non-significant | Significant |
| No-stenosis | **5** | 0 | 0 |
| Non-significant | 0 | **68** | 0 |
| Significant | 0 | 4 | **23** |

(Reference)

| Patient-based | Automatic | | |
|---|---|---|---|
| | No stenosis | Non-significant | Significant |
| No-stenosis | **2** | 0 | 0 |
| Non-significant | 0 | **13** | 0 |
| Significant | 0 | 2 | **16** |

(Reference)

stenosis or non-significant stenosis), the method achieved an accuracy of 0.93 and an F1 score of 0.83. For detection and determination of the anatomical significance of the stenosis (no stenosis, non-significant stenosis, significant stenosis), the method achieved an accuracy of 0.81, an average F1 score of 0.78 and a linearly weighted $\kappa$ of 0.70. At the artery-level, for detection of the significant stenosis the method achieved an accuracy of 0.96 and an F1 score of 0.95. For detection and determination of the anatomical significance of stenosis, the method achieved an accuracy of 0.96, an average F1 score of 0.96 and a linearly weighted $\kappa$ of 0.92. At the patient-level, for detection of the significant stenosis, the method achieved an accuracy of 0.94 and an F1 score of 0.94. For detection and determination of the anatomical significance of stenosis, the method achieved an accuracy of 0.94, an average F1 score of 0.96 and a linearly weighted $\kappa$ of 0.90.

*Table 3: Accuracy (Acc.), unweighted F1 score and Cohen's $\kappa$ at the fragment-level for plaque and stenosis characterization using three different network architectures. CNN-only stands for the network where the recurrent layers were replaced by fully connected layers. Single stands for the network with single softmax classifier. Note that, for plaque detection and characterization, the unweighted $\kappa$ was used, while for stenosis detection and determination of the significance of the detected stenosis, linearly weighted $\kappa$ was used.*

| | Plaque analysis | | | Stenosis analysis | | |
|---|---|---|---|---|---|---|
| | Acc. | F1 | $\kappa$ | Acc. | F1 | $\kappa$ |
| CNN-only | 0.65 | 0.61 | 0.49 | 0.75 | 0.74 | 0.60 |
| Single | 0.65 | 0.57 | 0.49 | 0.71 | 0.70 | 0.54 |
| **Proposed** | **0.72** | **0.61** | **0.60** | **0.81** | **0.78** | **0.70** |

To establish the value of the recurrent nature of proposed network, an additional experiment was performed in which a network with an identical CNN architecture was utilized, but the RNN was replaced with fully connected (FC) layers. To deal with different sequence lengths, and to aggregate the features resulting from the CNN analysis into one vector, a single global max pooling layer was used after the CNN. This layer was subsequently connected to two FC layers instead of the GRUs. To match the total number of trainable parameters in both architectures, the number of units in each of FC layers was raised from 64 to 192. In total, the network had 341,191 parameters (note that

RCNN network has 340,295 parameters). This architecture is illustrated in the Appendix. To allow a comparison with the proposed network, this network was trained and tested with the same training and test images. The obtained results are listed in Table 3.

Given that plaque and stenoses analyses are related, the classification could be posed as a single task with seven unique output classes (no plaque, calcified, mixed or non-calcified plaque without significant stenosis, calcified, mixed or non-calcified plaque with significant stenosis). To evaluate the performance of this merged classification, a network with identical architecture was utilized. However, the two softmax classifiers were replaced by a single softmax classifier with 7 output units. To allow comparison with the proposed network, this network was trained and tested with the same training and test images. The obtained results are listed in Table 3.

## 4.1 Comparison with previous work

Most published methods reported the fragment-based sensitivity and positive predictive value (PPV) for the detection of the anatomically significant stenosis (3). Shahzad et al. (12) reported a sensitivity of 0.55 and a PPV of 0.27, while Wang et al. (13) reported a sensitivity of 0.11 and a PPV of 0.33, and Broersen et al. (14) reported a sensitivity of 0.28 and a PPV 0.31. Using the same metrics, the proposed network achieved a sensitivity of 0.61 and a PPV of 0.83. Nonetheless, these methods reported performance using a different evaluation procedure and using different sets of patients than the current work. Therefore, a direct comparison of the results is not feasible and this can only be used as indication. Moreover, most methods for the detection of coronary artery plaque perform volume quantification of the calcified or non-calcified plaque. Therefore comparison with these methods is not feasible.

## 5 Discussion and conclusion

A method for automatic detection and characterization of the coronary artery plaque type, as well as detection and determination of the anatomical significance of the coronary artery stenosis was presented. First, features describing the volume along the coronary artery centerline were extracted by a 3D CNN. Subsequently, an RNN analyzed the extracted features to perform both classification tasks. Unlike most previous methods that detect and characterize coronary artery plaque and stenosis relying on the coronary artery lumen segmentation (3), the proposed method requires only the coronary artery centerline.

Anatomically significant stenoses could potentially lead to myocardial ischemia, and clinical guidelines suggest that different stenotic grades in coronary artery should be managed differently (2). Our experiments demonstrate that the proposed method was able to detect and determine the anatomical significance of coronary artery stenosis accurately with excellent reliability (Table 2). However, given that the population used in this study is diseased, the number of arteries and patients without any stenosis is low (Table 2). This might have biased our performance towards this population. Future work may address this by enlarging the data set with more images of patients with healthy arteries. Moreover, in this work, only two distinct degrees of stenosis were differentiated. Future work may introduce automatic determination of additional clinically relevant stenotic grades (e.g. <25% or >70%). For that, a larger training set of patients with manual annotations is required.

The results reveal that detection and characterization of coronary artery plaque type can be performed accurately, but with moderate reliability (Table 1). When evaluating the performance of the method for plaque detection only (plaque vs. no plaque), the results were accurate. However, further analysis of plaque characterization revealed that differentiation of the mixed plaque from the calcified and non-calcified plaque remains challenging. This is not surprising given that the mixed plaque contains both calcified and non-calcified components and that the distinction between mixed and calcified plaque, as well as between mixed and non-calcified plaque, is not clearly defined (see Appendix, Fig. 4). Alternatively, the automatic method could perform detection of calcified and non-calcified components only, and the obtained results could be merged into calcified, non-calcified and mixed plaque based on their spatial distribution. Moreover, ideally it would be interesting to segment plaque on a voxel basis, but obtaining voxel-wise reference is very labor intensive and requires an experienced expert.

Furthermore, the contribution of the recurrent nature of the proposed network was evaluated. The results show clear advantage of the proposed recurrent architecture over the network containing no recurrent units. This is in agreement with our assumption that a sequential analysis is needed to aggregate the knowledge of the entire analyzed region rather than just locally.

Unlike most methods for coronary artery plaque and stenosis classification that depend on the extracted artery centerline followed by arterial lumen segmentation (3), the proposed method relies only on the extracted artery centerline. Arterial lumen segmentation is far from trivial task, which occasionally requires substantial manual interaction, especially in diseased populations with heavily calcified arteries. We have here prevented potential error propagation by omitting this step altogether. To extract artery centerlines, we have used our method for artery centerline extraction (15), however, any other manual, semi-automatic or automatic method could be employed instead.

In this work, we have treated plaque and stenosis characterization as two different tasks, that were performed jointly. Although this halved the time of inference (1.8 seconds per artery), it has a limitation. Given that the parameters of the two softmax classifiers in the network differ, a physiologically impossible scenario, where a plaque is not detected while a stenosis is detected, can occur. Although in our experiments this was the case in only $< 1.5\%$ of the cases, future work should address this either by modifying the network architecture preventing such scenario, or by applying a high penalty for such cases in the loss of the network. Moreover, experiments comparing the proposed multi-task network with a network with single classifier demonstrate superior performance for the multi-task approach. The limited number of training samples may have prevented the single-task network from generalizing well with a relatively small network. In future work, size of the dataset may be increased by obtaining manual annotations in more images as well as by utilizing additional data augmentation methods.

The nature of reference annotations used in this work has limitations. Manual annotation assigned a single label to a whole fragment of the coronary artery containing plaque. Separating these fragments to their local components might lead to different labels. Consequently, the identified start and end points of the automatically detected plaque and stenosis are not in full agreement with the reference annotations (see Appendix, Fig. 3 and Fig. 4). Future work might address these limitations by modifying the reference standard so that each voxel in the arterial wall or each cross-section of the arterial lumen is annotated.

To conclude, this study presented an algorithm, based on a recurrent convolutional neural network, for automatic detection and characterization of coronary artery plaque, as well as detection and determination of the anatomical significance of coronary artery stenosis. This may enable automated triage of patients to those without coronary plaque, and those with coronary plaque and stenosis in need for further cardiovascular workup.

### Acknowledgments

This study was financially supported by the project FSCAD, funded by the Netherlands Organization for Health Research and Development (ZonMw) in the framework of the research programme IMDI (Innovative Medical Devices Initiative); project 104003009.

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

# Appendix

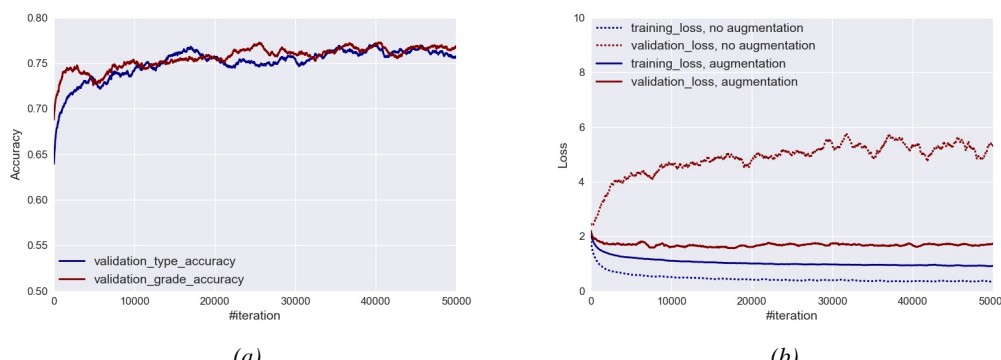

*(a)*            *(b)*

*Figure 1: (a) Validation accuracies for detection and characterization of coronary artery plaque, as well as detection and determination of the anatomical significance of coronary artery stenosis during training. (b) Training and validation losses during training with and without data augmentation.*

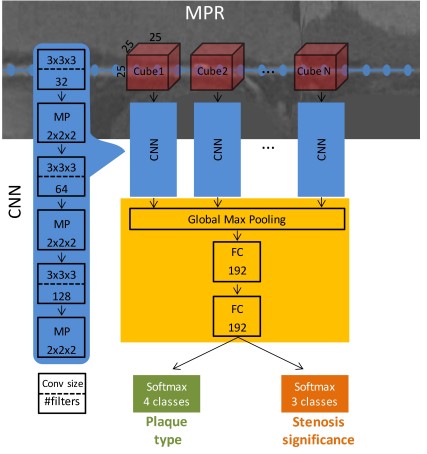

*Figure 2: The input of the network is a sequence of cubes extracted from the MPR, along artery centerline. A CNN extracts features out of each $25 \times 25 \times 25$ voxels cube, then a global max pooling and two dense layers process the entire sequence. The output of the RNN is fed into two softmax classifiers to simultaneously characterize plaque and stenosis. To match the total number of parameters compared to the proposed network, 196 units in each of FC layers were used. In total, the network had 341,191 parameters, vs. 340,295 parameters in proposed network.*

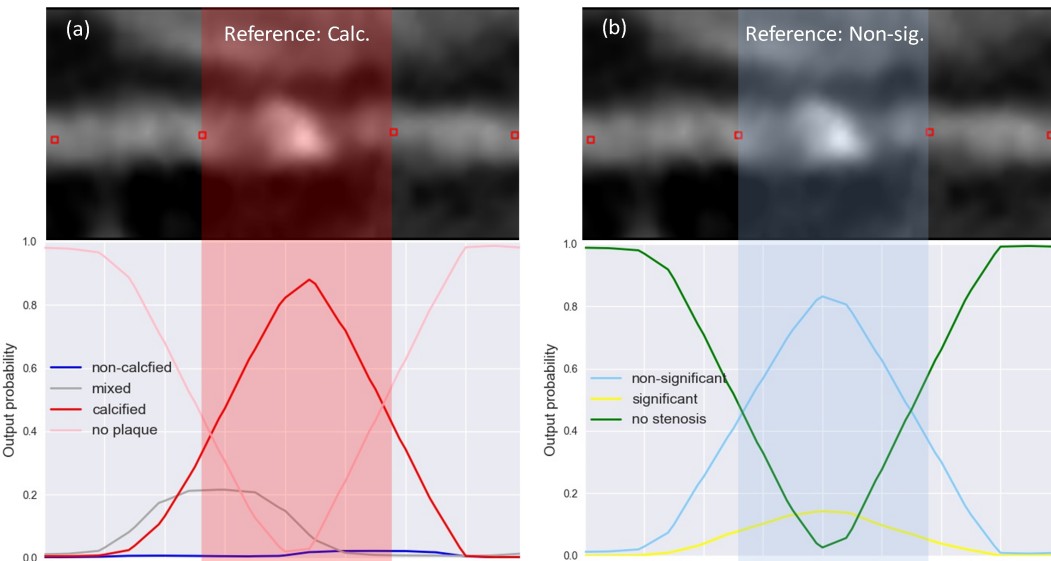

*Figure 3: An example of output probabilities for both softmax classifiers. (a) MPR view of an artery with reference standard for plaque type with a manually annotated single fragment, its boundaries and the reference label (top), and the output probabilities for plaque detection and characterization (bottom). (b) MPR view of an artery with reference standard for stenosis significance with a manually annotated single fragment, its boundaries and the reference label (top), and the output probabilities for stenosis detection and determination of its significance (bottom). Non-Sig. = Non-significant stenosis. Calc. = Calcified plaque. Non-calc. = Non-calcified plaque.)*

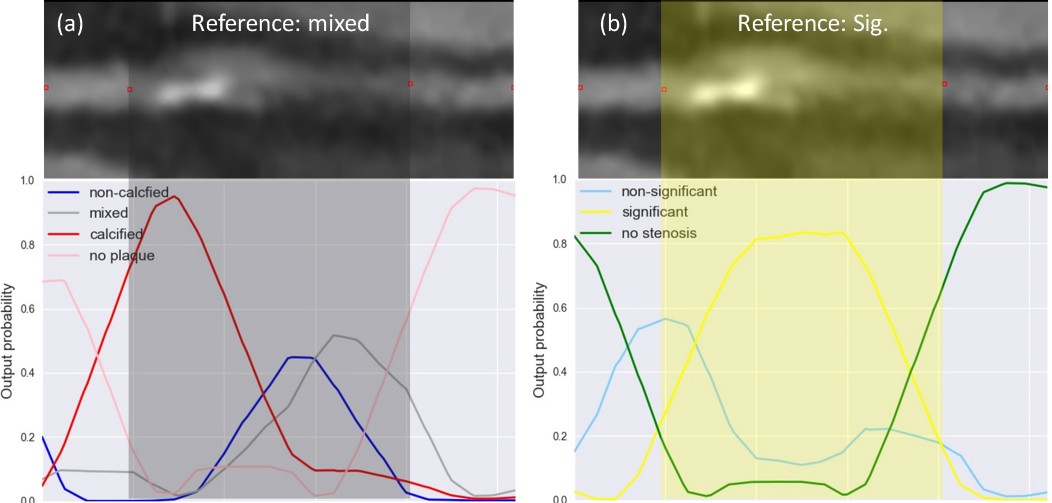

*Figure 4: An example of output probabilities for both softmax classifiers. (a) MPR view of an artery with reference standard for plaque type with a manually annotated single fragment, its boundaries and the reference label (top), and the output probabilities for plaque detection and characterization (bottom). (b) MPR view of an artery with reference standard for stenosis significance with a manually annotated single fragment, its boundaries and the reference label (top), and the output probabilities for stenosis detection and determination of its significance (bottom). Sig. = Significant stenosis.)*

