# OpenReview forum: "Automatic Detection and Characterization of Coronary Artery Plaque and Stenosis using a Recurrent Convolutional Neural Network in Coronary CT Angiography"
_MIDL.amsterdam/2018/Conference — MIDL 2018 Poster_

### Review · AnonReviewer3 · 2018-05-08
**Novel contribution to the analysis of CCTA**

**Rating:** 4
**Confidence:** 2

**Review:**

This work applied a recurrent convolutional neural network to CCTA images to detect and characterize coronary artery plaque, as well as to detect and determine the significance of stenosis. Results were reported on a proprietary dataset, and yielded promising results. This work exploited the sequential nature of CCTA data, and thoroughly addressed the limitations of the current model. It would be a novel contribution to the CCTA literature.

Some specific notes:
- Good literature review
- Some grammar corrections required
- Data was labelled by only one expert - having the data labelled by another expert could yield more reliable training labels, potentially improving the performance metrics of the model. As mentioned in the introduction, the manual analysis of these images is subject to inter-observer variability, so perhaps taking the agreement of multiple observers into account would be beneficial.
- Were all arteries included in this study? (i.e. RCA, LAD, CA) Or did it only focus one type? This should be mentioned somewhere in-text. An analysis of performance per artery type could be interesting.
- Evaluation Section (3.4) - this section is a little jumbled and should be clarified and made more concise and clear
- In the Results, F1 scores and k-metrics are all in-text -- make a table for better comparisons (like Table 3)
- Table 3 appears before it is discussed in-text
- Good analysis of multiple model configurations
- Why not also test on the publicly available dataset mentioned in the Introduction? (http://coronary.bigr.nl/) From what is available, it looks like the labels could be used for the stenosis significance task. This would allow for benchmark comparisons, and would add weight to this work.
- Good address of current model limitations in the Discussion

This work should be accepted, but with significant revisions to grammar and clarity, as well as re-organization and clarification of the results section.

**Special Issue:**

No

---

### Review · AnonReviewer2 · 2018-05-14
**seems to work well, some experiments required to tell us why (and whether it's better than competing architectures)**

**Rating:** 3
**Confidence:** 2

**Review:**

Summary:

The authors use RNN CNN to classify vessel calcifications in CCTA, report interesting results

Pro:

- This appears to be a first CNN paper on an intersting task / data set
- The authors propose a sequential processing of images acquired along the vessel. This is an interesting direction of using RNN.


Con:

- it would have been interesting to see the performance of naive competing architectures, e.g., a Unet like segmentation.
- it would be be interesting to see to what extend the "temporal" direction of the RNN is relevant over a plain volumetric processing (that would - eventually - also be applicable without prior centerline extraction).
- Fig 2 is incomprehensible due to text size (at least in my pdf)


**Special Issue:**

Yes

---

### Review · AnonReviewer1 · 2018-05-17

**Rating:** 4
**Confidence:** 2

**Review:**

The paper uses an RCNN for detecting and characterising coronary plaque in CCTA. Data from 131 patients was used. Detection and characterisation was performed in two steps. The RCNN uses a sequence of 25-cubed data block along the artery centerline. The conclusion is the the RNN structure improves performance.

**Special Issue:**

No

---

### Decision · Program_Chairs · 2018-05-15
**Paper72 Acceptance Decision**

Poster